# A Fast and Provable Algorithm for Sparse Phase Retrieval

## Abstract

We study the sparse phase retrieval problem, which aims to recover a sparse signal from a limited number of phaseless measurements. Existing algorithms for sparse phase retrieval primarily rely on first-order methods with linear convergence rate. In this paper, we propose an efficient second-order algorithm based on Newton projection, which maintains the same per-iteration computational complexity as popular first-order methods. The proposed algorithm is theoretically guaranteed to converge to the ground truth (up to a global sign) at a quadratic convergence rate after at most $\mathcal{O}\big(\log(\|\boldsymbol{x}^{\natural}\|/x^{\natural}_{\min})\big)$ iterations, provided a sample complexity of $\mathcal{O}(s^2 \log n)$, where $\boldsymbol{x}^{\natural} \in \mathbb{R}^n$ represents an $s$-sparse ground truth signal. Numerical experiments demonstrate that our algorithm not only outperforms state-of-the-art methods in terms of achieving a significantly faster convergence rate, but also excels in attaining a higher success rate for exact signal recovery from noise-free measurements and providing enhanced signal reconstruction in noisy scenarios.

## 1 Introduction

We study the phase retrieval problem, which involves reconstructing an $n$-dimensional signal $\boldsymbol{x}^{\natural}$ using its intensity-only measurements:

$$y_i = |\langle \boldsymbol{a}_i, \boldsymbol{x}^{\natural}\rangle|^2, \quad i = 1, 2, \cdots, m, \tag{1}$$

where each $y_i$ represents a measurement, $\boldsymbol{a}_i$ denotes a sensing vector, $\boldsymbol{x}^{\natural}$ is the unknown signal to be recovered, and $m$ is the total number of measurements. The phase retrieval problem arises in various applications, including diffraction imaging [1], X-ray crystallography [2, 3], and optics [4], where detectors can only record the squared modulus of Fresnel or Fraunhofer diffraction patterns of radiation scattered from an object. The loss of phase information complicates the understanding of the scattered object, as much of the image's structural content may be encoded in the phase.

Although the phase retrieval problem is ill-posed and even NP-hard [5], several algorithms have been proven to succeed in recovering target signals under certain assumptions. Algorithms can be broadly categorized into convex and nonconvex approaches. Convex methods, such as PhaseLift [6, 7], PhaseCut [8], and PhaseMax [9, 10], offer optimal sample complexity but are computationally challenging in high-dimensional cases. To improve computational efficiency, nonconvex approaches are explored, including alternating minimization [11], Wirtinger flow [6], truncated amplitude flow [12], Riemannian optimization [13], Gauss-Newton [14, 15], and Kaczmarz [16, 17]. Despite the nonconvex nature of its objective function, the global geometric landscape lacks spurious local minima [18, 19], allowing algorithms with random initialization to work effectively [20, 21].

Submitted to 37th Conference on Neural Information Processing Systems (NeurIPS 2023). Do not distribute.

The nonconvex approaches previously mentioned can guarantee successful recovery of the ground truth (up to a global phase) with a sample complexity $m \sim \mathcal{O}(n \log^a n)$, where $a \geq 0$. This complexity is nearly optimal, as the phase retrieval problem requires $m \geq 2n - 1$ for real signals and $m \geq 4n - 4$ for complex signals [22]. However, in practical situations, especially in high-dimensional cases, the number of available measurements is often less than the signal dimension (*i.e.*, $m < n$), leading to a need for further reduction in sample complexity.

In this paper, we focus on the sparse phase retrieval problem, which aims to recover a sparse signal from a limited number of phaseless measurements. It has been established that the minimal sample complexity required to ensure $s$-sparse phase retrievability in the real case is only $2s$ for generic sensing vectors [23]. Several algorithms have been proposed to address the sparse phase retrieval problem [24, 25, 26, 27, 28]. These approaches have been demonstrated to effectively reconstruct the ground truth using $\mathcal{O}(s^2 \log n)$ Gaussian measurements. While this complexity is not optimal, it is significantly smaller than that in general phase retrieval.

## 1.1 Contributions

Existing algorithms for sparse phase retrieval primarily employ first-order methods with linear convergence. Recent work [28] introduced a second-order method, while it fails to obtain a quadratic convergence rate. The main contributions of this paper can be summarized in three key points:

1. We propose a second-order algorithm based on Newton projection for sparse phase retrieval that maintains the same per-iteration computational complexity as popular first-order methods. To ensure fast convergence, we integrate second-order derivative information from intensity-based empirical loss into the search direction; to ensure computational efficiency, we restrict the Newton update to a subset of variables, setting others to zero in each iteration.

2. We establish a non-asymptotic quadratic convergence rate for our proposed algorithm and provide the iteration complexity. Specifically, we prove that the algorithm converges to the ground truth (up to a global sign) at a quadratic rate after at most $\mathcal{O}\left(\log(|\boldsymbol{x}^\natural| / x_{\min}^\natural)\right)$ iterations, provided a sample complexity of $\mathcal{O}(s^2 \log n)$. To the best of our knowledge, this is the first algorithm to establish a quadratic convergence rate for sparse phase retrieval.

3. Numerical experiments demonstrate that the proposed algorithm achieves a significantly faster convergence rate in comparison to state-of-the-art methods. Furthermore, the experiments reveal that our algorithm attains a higher success rate in exact signal recovery from noise-free measurements and provides enhanced signal reconstruction performance in noisy scenarios, as evidenced by the improved Peak Signal-to-Noise Ratio (PSNR).

**Notation:** The $p$-norm $\|\boldsymbol{x}\|_p := \left(\sum_{i=1}^n |x_i|^p\right)^{1/p}$ for $p \geq 1$. $\|\boldsymbol{x}\|_0$ denotes the number of nonzero entries of $\boldsymbol{x}$, and $\|\boldsymbol{x}\|$ denotes the 2-norm. For a matrix $\boldsymbol{A} \in \mathbb{R}^{m \times n}$, $\|\boldsymbol{A}\|$ is the spectral norm of $\boldsymbol{A}$. For any $q_1 \geq 1$ and $q_2 \geq 1$, $\|\boldsymbol{A}\|_{q_2 \to q_1}$ denotes the induced operator norm from the Banach space $(\mathbb{R}^n, \|\cdot\|_{q_2})$ to $(\mathbb{R}^m, \|\cdot\|_{q_1})$. $\lambda_{\min}(\boldsymbol{A})$ and $\lambda_{\max}(\boldsymbol{A})$ denote the smallest and largest eigenvalues of the matrix $\boldsymbol{A}$. $|\mathcal{S}|$ denotes the number of elements in $S$. $\boldsymbol{a} \odot \boldsymbol{b}$ denotes the entrywise product of $\boldsymbol{a}$ and $\boldsymbol{b}$. For functions $f(n)$ and $g(n)$, we write $f(n) \lesssim g(n)$ if $f(n) \leq cg(n)$ for some constant $c \in (0, +\infty)$. For $\boldsymbol{x}, \boldsymbol{x}^\natural \in \mathbb{R}^n$, the distance between $\boldsymbol{x}$ and $\boldsymbol{x}^\natural$ is defined as $\text{dist}(\boldsymbol{x}, \boldsymbol{x}^\natural) := \min\left\{\|\boldsymbol{x} - \boldsymbol{x}^\natural\|, \|\boldsymbol{x} + \boldsymbol{x}^\natural\|\right\}$. $x_{\min}^\natural$ denotes the smallest nonzero entry in magnitude of $\boldsymbol{x}^\natural$.

## 2 Problem Formulation and Related Works

We first present the problem formulation for sparse phase retrieval, and then review related works.

### 2.1 Problem formulation

The standard sparse phase retrieval problem can be concisely expressed as finding $\boldsymbol{x}$ that satisfies

$$|\langle \boldsymbol{a}_i, \boldsymbol{x} \rangle|^2 = y_i \quad \forall i = 1, \ldots, m, \quad \text{and} \quad \|\boldsymbol{x}\|_0 \leq s, \tag{2}$$

where $\{\boldsymbol{a}_i\}_{i=1}^m$ are known sensing vectors and $\{y_i\}_{i=1}^m$ represent phaseless measurements with $y_i = |\langle \boldsymbol{a}_i, \boldsymbol{x}^\natural \rangle|^2$, where $\boldsymbol{x}^\natural$ is the ground truth signal ($\|\boldsymbol{x}^\natural\|_0 \leq s$). While sparsity level $s$ is assumed known a priori for theoretical analysis, our experiments will also explore cases with unknown $s$.

Table 1: Overview of per-iteration computational cost, numbers of iterations for convergence, loss function, and algorithm types for various methods. $\boldsymbol{x}^{\natural}$ represents the ground truth signal with dimension $n$ and sparsity $s$, and $x^{\natural}_{\min}$ denotes the smallest nonzero entry in magnitude of $\boldsymbol{x}^{\natural}$.

| Methods | Per-iteration cost | Iteration complexity | Loss function | Algorithm types |
|---|---|---|---|---|
| ThWF [25] | $\mathcal{O}(n^2 \log n)$ | $\mathcal{O}(\log(1/\epsilon))$ | $f_I(\boldsymbol{x})$ | Grad. Proj. |
| SPARTA [26] | $\mathcal{O}(ns^2 \log n)$ | $\mathcal{O}(\log(1/\epsilon))$ | $f_A(\boldsymbol{x})$ | Grad. Proj. |
| CoPRAM [27] | $\mathcal{O}(ns^2 \log n)$ | $\mathcal{O}(\log(1/\epsilon))$ | $f_A(\boldsymbol{x})$ | Alt. Min. |
| HTP [28] | $\mathcal{O}((n + s^2)s^2 \log n)$ | $\mathcal{O}(\log(s^2 \log n) + \log(\|\boldsymbol{x}^{\natural}\|/x^{\natural}_{\min}))$ | $f_A(\boldsymbol{x})$ | Alt. Min. |
| Proposed | $\mathcal{O}((n + s^2)s^2 \log n)$ | $\mathcal{O}(\log(\log(1/\epsilon)) + \log(\|\boldsymbol{x}^{\natural}\|/x^{\natural}_{\min}))$ | $f_I(\boldsymbol{x})$ | Newton Proj. |

To address Problem (2), various problem reformulations have been explored. Convex formulations, such as the $\ell_1$-regularized PhaseLift method [24], often use the lifting technique and solve the problem in the $n \times n$ matrix space, resulting in high computational costs. To enhance computational efficiency, nonconvex approaches [25, 26, 28, 29] are explored, which can be formulated as:

$$\underset{\boldsymbol{x}}{\text{minimize}}\ f(\boldsymbol{x}), \qquad \text{subject to} \quad \|\boldsymbol{x}\|_0 \le s. \tag{3}$$

Both the loss function $f(\boldsymbol{x})$ and the $\ell_0$-norm constraint in Problem (3) are nonconvex, making it challenging to solve. Two prevalent loss functions are investigated: intensity-based empirical loss

$$f_I(\boldsymbol{x}) := \frac{1}{4m} \sum_{i=1}^{m} \left( |\langle \boldsymbol{a}_i, \boldsymbol{x} \rangle|^2 - y_i \right)^2, \tag{4}$$

and amplitude-based empirical loss

$$f_A(\boldsymbol{x}) := \frac{1}{2m} \sum_{i=1}^{m} \left( |\langle \boldsymbol{a}_i, \boldsymbol{x} \rangle| - z_i \right)^2, \tag{5}$$

where $z_i = \sqrt{y_i}$, $i = 1, \ldots, m$. The intensity-based loss $f_I(\boldsymbol{x})$ is smooth, while the amplitude-based loss $f_A(\boldsymbol{x})$ is non-smooth because of the modulus.

## 2.2 Related works

Existing nonconvex sparse phase retrieval algorithms can be broadly classified into two categories: gradient projection methods and alternating minimization methods. Gradient projection methods, such as ThWF [25] and SPARTA [26], employ thresholded gradient descent and iterative hard thresholding, respectively. On the other hand, alternating minimization methods, including CoPRAM [27] and HTP [28], alternate between updating the signal and phase. When updating the signal, formulated as a sparsity-constrained least squares problem, CoPRAM leverages the cosamp method [30], while HTP applies the hard thresholding pursuit algorithm [31]. In this paper, we introduce a Newton projection-based algorithm that incorporates second-order derivative information, resulting in a faster convergence rate compared to gradient projection methods, and, unlike alternating minimization methods, it eliminates the need for separate signal and phase updates. We note that ThWF and our algorithm utilize intensity-based loss as the objective function, while SPARTA, CoPRAM, and HTP employ amplitude-based loss. All these algorithms require a sample complexity of $\mathcal{O}(s^2 \log n)$ under Gaussian measurements for successful recovery.

The majority of sparse phase retrieval algorithms, such as ThWF, SPARTA, and CoPRAM, are first-order methods with linear convergence rates. While HTP is a second-order method that converges in a finite number of iterations, it fails to establish a quadratic convergence rate. We propose a second-order algorithm that attains a non-asymptotic quadratic convergence rate and exhibits lower iteration complexity compared to HTP. Our algorithm maintains the same computational complexity per iteration as popular first-order methods when $s \lesssim \sqrt{n}$. This condition is always assumed to hold true; otherwise, the established sample complexity for sparse phase retrieval algorithms, $\mathcal{O}(s^2 \log n)$, would be reduced to that of general phase retrieval methods. Table 1 presents a comparative overview of the previously discussed methods and our proposed method.

## 3 Main Results

In this section, we present our proposed algorithm for sparse phase retrieval. Generally, nonconvex methods comprise two stages: initialization and refinement. The first stage generates an initial guess close to the target signal, while the second stage refines the initial guess using various methods, such as ThWF, SPARTA, CoPRAM, and HTP. Our proposed algorithm adheres to this two-stage strategy. In the first stage, we employ an existing effective method to generate an initial point. Our primary focus is on the second stage, wherein we propose an efficient second-order algorithm based on Newton projection to refine the initial guess.

Before delving into the details of our proposed algorithm, we present a unified algorithmic framework for addressing the sparsity-constrained optimization problem in Eq. (3), as summarized in [32]. Given the $k$-th iterate $\boldsymbol{x}^k$, the next iterate $\boldsymbol{x}^{k+1}$ can be obtained through the following steps:

**Step 1** (Hard thresholding):

$$\boldsymbol{u}^{k+1} = \mathcal{H}_r(\phi(\boldsymbol{x}^k)), \tag{6}$$

**Step 2** (Debiasing):

$$\boldsymbol{v}^{k+1} = \arg\min_{\boldsymbol{x}} \psi_k(\boldsymbol{x}), \quad \text{subject to} \quad \text{supp}(\boldsymbol{x}) \subseteq \mathcal{S}_{k+1}, \tag{7}$$

**Step 3** (Pruning):

$$\boldsymbol{x}^{k+1} \in \mathcal{H}_s(\boldsymbol{v}^{k+1}), \tag{8}$$

where $\phi(\boldsymbol{x}^k)$ is typically chosen as either $\nabla f(\boldsymbol{x}^k)$ or $\boldsymbol{x}^k - \eta \nabla f(\boldsymbol{x}^k)$, $\psi_k(\boldsymbol{x})$ is designed based on the objective function $f(\boldsymbol{x})$ and the iterate $\boldsymbol{x}^k$, and $\mathcal{S}_{k+1}$ is usually defined as the support of $\boldsymbol{u}^{k+1}$. The hard-thresholding operator, denoted by $\mathcal{H}_s$, is defined with a sparsity level of $s$ as follows:

$$\mathcal{H}_s(\boldsymbol{w}) := \arg\min_{\boldsymbol{x}} \|\boldsymbol{x} - \boldsymbol{w}\|^2, \quad \text{subject to} \quad \|\boldsymbol{x}\|_0 \leq s. \tag{9}$$

A variety of well-known algorithms for solving sparsity-constrained optimization problems adhere to the three-step algorithmic framework mentioned earlier. For instance, the Iterative Hard Thresholding (IHT) algorithm solely performs Step 1 using $\mathcal{H}_s(\boldsymbol{x}^k - \eta \nabla f(\boldsymbol{x}^k))$ with $\eta$ the stepsize; the Hard Thresholding Pursuit (HTP) implements the first two steps by computing $\boldsymbol{u}^{k+1}$ via one-step IHT in Step 1, and then solving the support-constrained problem in Step 2 with $\mathcal{S}_{k+1} = \text{supp}(\boldsymbol{u}^{k+1})$; the Compressive Sampling Matching Pursuit (CoSaMP) executes all three steps, calculating $\boldsymbol{u}^{k+1} = \mathcal{H}_{2s}(\nabla f(\boldsymbol{x}^k))$ in Step 1, performing Step 2 with $\mathcal{S}_{k+1} = \text{supp}(\boldsymbol{u}^{k+1}) \cup \text{supp}(\boldsymbol{x}^k)$, and pruning the result in Step 3 to ensure an $s$-sparse level.

Several state-of-the-art methods for sparse phase retrieval share strong connections with the previously described popular algorithms for sparsity-constrained optimization, and thus relate closely to the algorithmic framework. SPARTA combines IHT with gradient truncation to eliminate erroneously estimated signs. HTP merges hard thresholding pursuit with alternating minimization, updating the signal and phase alternately. CoPRAM integrates CoSaMP with alternating minimization. Our proposed algorithm will also be presented using this algorithmic framework.

### 3.1 Proposed algorithm

In this subsection, we introduce our proposed algorithm for sparse phase retrieval, which utilizes the intensity-based loss $f_I$ defined in Eq. (4) as the objective function. The algorithm incorporates the first two steps of the previously discussed algorithmic framework.

Our algorithm is developed based on the Newton projection method. It is worth mentioning that Newton-type methods typically require solving a linear system at each iteration to determine the Newton direction. This generally results in a computational cost of $\mathcal{O}(n^3)$ for our problem, rendering it impractical in high-dimensional situations. To address this challenge, we categorize variables into two groups at each iteration: *free* and *fixed*, updating them separately. The *free* variables, consisting of at most $s$ variables, are updated according to the (approximate) Newton direction, while the *fixed* variables are set to zero. This strategy requires solving a linear system of size $s \times s$, substantially decreasing the computational expense from $\mathcal{O}(n^3)$ to $\mathcal{O}(s^3)$.

In the first step, we identify the set of *free* variables using one-step IHT of the loss $f_A(\boldsymbol{x})$ in (5):
$$\mathcal{S}_{k+1} = \text{supp}\left(\mathcal{H}_s(\boldsymbol{x}^k - \eta \nabla f_A(\boldsymbol{x}^k))\right),$$
where $\eta$ is the stepsize. Since $f_A$ is non-smooth, we adopt the generalized gradient [33] as $\nabla f_A$. The $s$-sparse hard thresholding limits $|\mathcal{S}_{k+1}|$ to $s$, implying that there are at most $s$ *free* variables. We only update *free* variables along the approximate Newton direction and set others to zero.

In the second step, we update the *free* variables in $\mathcal{S}_{k+1}$ by solving a support-constrained problem in Eq. (7). Note that we adopt the intensity-based loss $f_I$ as the objective function. To accelerate convergence, we choose function $\psi_k(\boldsymbol{x})$ in (7) as the second-order Taylor expansion of $f_I$ at $\boldsymbol{x}^k$:

$$\psi_k(\boldsymbol{x}) := f_I(\boldsymbol{x}^k) + \langle \nabla f_I(\boldsymbol{x}^k),\ \boldsymbol{x} - \boldsymbol{x}^k \rangle + \frac{1}{2}\langle \boldsymbol{x} - \boldsymbol{x}^k,\ \nabla^2 f_I(\boldsymbol{x}^k)(\boldsymbol{x} - \boldsymbol{x}^k) \rangle.$$

Let $\boldsymbol{x}^\star$ denote the minimizer of Problem (7). For notational simplicity, define $\boldsymbol{g}_{\mathcal{S}_{k+1}}^k = \left[\nabla f_I(\boldsymbol{x}^k)\right]_{\mathcal{S}_{k+1}}$, which denotes the sub-vector of $\nabla f_I(\boldsymbol{x}^k)$ indexed by $\mathcal{S}_{k+1}$, $\boldsymbol{H}_{\mathcal{S}_{k+1}}^k = \left[\nabla^2 f_I(\boldsymbol{x}^k)\right]_{\mathcal{S}_{k+1}}$, which represents the principle sub-matrix of the Hessian indexed by $\mathcal{S}_{k+1}$, and $\boldsymbol{H}_{\mathcal{S}_{k+1},\mathcal{S}_{k+1}^c}^k = \left[\nabla^2 f_I(\boldsymbol{x}^k)\right]_{\mathcal{S}_{k+1},\mathcal{S}_{k+1}^c}$, denoting the sub-matrix of the Hessian whose rows and columns are indexed by $\mathcal{S}_{k+1}$ and $\mathcal{S}_{k+1}^c$, respectively. Following from the first-order optimality condition of Problem (7), we obtain that $\boldsymbol{x}_{\mathcal{S}_{k+1}^c}^\star = \boldsymbol{0}$ and $\boldsymbol{x}_{\mathcal{S}_{k+1}}^\star$ satisfies

$$\boldsymbol{H}_{\mathcal{S}_{k+1}}^k\left(\boldsymbol{x}_{\mathcal{S}_{k+1}}^\star - \boldsymbol{x}_{\mathcal{S}_{k+1}}^k\right) = \boldsymbol{H}_{\mathcal{S}_{k+1},\mathcal{S}_{k+1}^c}^k \boldsymbol{x}_{\mathcal{S}_{k+1}^c}^k - \boldsymbol{g}_{\mathcal{S}_{k+1}}^k. \tag{10}$$

As a result, we obtain the next iterate $\boldsymbol{x}^{k+1}$ by
$$\boldsymbol{x}_{\mathcal{S}_{k+1}}^{k+1} = \boldsymbol{x}_{\mathcal{S}_{k+1}}^k - \boldsymbol{p}_{\mathcal{S}_{k+1}}^k, \quad \text{and} \quad \boldsymbol{x}_{\mathcal{S}_{k+1}^c}^{k+1} = \boldsymbol{0}, \tag{11}$$

where $\boldsymbol{p}_{\mathcal{S}_{k+1}}^k$ represents the approximate Newton direction over $\mathcal{S}_{k+1}$, which can be calculated by
$$\boldsymbol{H}_{\mathcal{S}_{k+1}}^k \boldsymbol{p}_{\mathcal{S}_{k+1}}^k = -\boldsymbol{H}_{\mathcal{S}_{k+1},J_{k+1}}^k \boldsymbol{x}_{J_{k+1}}^k + \boldsymbol{g}_{\mathcal{S}_{k+1}}^k. \tag{12}$$

where $J_{k+1} := \mathcal{S}_k \setminus \mathcal{S}_{k+1}$ with $|J_{k+1}| \leq s$. In contrast to Eq. (10), we replace $\boldsymbol{x}_{\mathcal{S}_{k+1}^c}^k$ with $\boldsymbol{x}_{J_{k+1}}^k$ in (12), as $J_{k+1}$ captures all nonzero elements in $\boldsymbol{x}_{\mathcal{S}_{k+1}^c}^k$ as follows:

$$\mathcal{G}\left(\boldsymbol{x}_{\mathcal{S}_{k+1}^c}^k\right) = \begin{bmatrix} \boldsymbol{x}_{\mathcal{S}_{k+1}^c \cap \mathcal{S}_k}^k \\ \boldsymbol{0} \end{bmatrix} = \begin{bmatrix} \boldsymbol{x}_{\mathcal{S}_k \setminus \mathcal{S}_{k+1}}^k \\ \boldsymbol{0} \end{bmatrix} = \begin{bmatrix} \boldsymbol{x}_{J_{k+1}}^k \\ \boldsymbol{0} \end{bmatrix}, \tag{13}$$

where operator $\mathcal{G}$ arranges all nonzero elements of a vector to appear first, followed by zero elements. The first equality in (13) follows from the fact that $\text{supp}(\boldsymbol{x}^k) \subseteq \mathcal{S}_k$. By calculating $\boldsymbol{H}_{\mathcal{S}_{k+1},J_{k+1}}^k$ rather than $\boldsymbol{H}_{\mathcal{S}_{k+1},\mathcal{S}_{k+1}^c}^k$ as in (12), the computational cost is substantially reduced from $\mathcal{O}(smn)$ to $\mathcal{O}(s^2m)$. The costs for computing $\boldsymbol{H}_{\mathcal{S}_{k+1}}^k$ and solving the linear system in (12) are $\mathcal{O}(s^2m)$ and $\mathcal{O}(s^3)$, respectively. Therefore, the overall computational cost for Step 2 is $\mathcal{O}(s^2m)$, while the cost for Step 1 amounts to $\mathcal{O}(mn)$, which involves calculating $\nabla f_A(\boldsymbol{x}^k)$.

In summary, the computational costs for Steps 1 and 2 are $\mathcal{O}(mn)$ and $\mathcal{O}(s^2m)$, respectively, making the total cost per iteration $\mathcal{O}(n + s^2)m$, with $m \sim \mathcal{O}(s^2 \log n)$ that is required for successful recovery. Since $s \lesssim \sqrt{n}$ is always assumed to hold true as discussed in Section 2.2, the per-iteration computational complexity of our algorithm is equivalent to that of popular first-order methods, which is $\mathcal{O}(ns^2 \log n)$. The pruning step is omitted as $\boldsymbol{x}^{k+1}$ in (11) is already $s$-sparse.

---

**Algorithm 1** Proposed algorithm

---

**Input:** Data $\{\boldsymbol{a}_i, y_i\}_{i=1}^m$, sparsity $s$, initial estimate $\boldsymbol{x}^0$, and stepsize $\eta$.

1: **for** $k = 0, 1, 2, \ldots$ **do**

2:     Identify the set of *free* variables $\mathcal{S}_{k+1} = \text{supp}(\mathcal{H}_s(\boldsymbol{x}^k - \eta \nabla f_A(\boldsymbol{x}^k)))$;

3:     Compute the approximate Newton direction $\boldsymbol{p}_{\mathcal{S}_{k+1}}^k$ over $\mathcal{S}_{k+1}$ by solving (12).

4:     Update $\boldsymbol{x}^{k+1}$:
$$\boldsymbol{x}_{\mathcal{S}_{k+1}}^{k+1} = \boldsymbol{x}_{\mathcal{S}_{k+1}}^k - \boldsymbol{p}_{\mathcal{S}_{k+1}}^k, \quad \text{and} \quad \boldsymbol{x}_{\mathcal{S}_{k+1}^c}^{k+1} = \boldsymbol{0}.$$

5: **end for**

**Output:** $\boldsymbol{x}^{k+1}$.

---

## 3.2 Initialization

The nonconvex nature of phase retrieval problems often requires a well-designed initial guess to find a global minimizer. Spectral initialization is a common approach [6]. In this paper, we adopt a sparse variant of the spectral initialization method to obtain a favorable initial guess for Algorithm 1.

Assuming $\{\boldsymbol{a}_i\}_{i=1}^m$ are independently drawn from a Gaussian distribution $\mathcal{N}(\boldsymbol{0}, \boldsymbol{I}_n)$, the expectation of the matrix $\frac{1}{m}\sum_{i=1}^m y_i \boldsymbol{a}_i \boldsymbol{a}_i^T$ is $\boldsymbol{M} := \|\boldsymbol{x}^\natural\|^2 \boldsymbol{I}_n + 2\boldsymbol{x}^\natural (\boldsymbol{x}^\natural)^T$. The leading eigenvector of $\boldsymbol{M}$ is precisely $\pm \boldsymbol{x}^\natural$. Hence, the leading eigenvector of $\frac{1}{m}\sum_{i=1}^m y_i \boldsymbol{a}_i \boldsymbol{a}_i^T$ can be close to $\pm\boldsymbol{x}^\natural$ [6]. However, this method requires the sample complexity of at least $\mathcal{O}(n)$, which is excessively high for sparse phase retrieval. Leveraging the sparsity of $\boldsymbol{x}^\natural$ is crucial to lower this complexity.

We adopt the sparse spectral initialization method proposed in [27]. Specifically, we first collect the indices of the largest $s$ values from $\left\{\frac{1}{m}\sum_{i=1}^m y_i [\boldsymbol{a}_i]_j^2\right\}_{j=1}^n$ and obtain the set $\hat{S}$, which serves as an estimate of the support of the true signal $\boldsymbol{x}^\natural$. Next, we construct the initial guess $\boldsymbol{x}^0$ as follows: $\boldsymbol{x}^0_{\hat{S}}$ is the leading eigenvector of $\frac{1}{m}\sum_{i=1}^m y_i [\boldsymbol{a}_i]_{\hat{S}} [\boldsymbol{a}_i]_{\hat{S}}^T$, and $\boldsymbol{x}^0_{\hat{S}^c} = \boldsymbol{0}$. Finally, we scale $\boldsymbol{x}^0$ such that $\|\boldsymbol{x}^0\|^2 = \frac{1}{m}\sum_{i=1}^m y_i$, ensuring the power of $\boldsymbol{x}^0$ closely aligns with the power of $\boldsymbol{x}^\natural$.

The study in [27] demonstrates that, given a sample complexity $m \sim \mathcal{O}(s^2 \log n)$, the aforementioned sparse spectral initialization method can produce an initial estimate $\boldsymbol{x}^0$ that is sufficiently close to the ground truth. Specifically, it holds $\mathrm{dist}(\boldsymbol{x}^0, \boldsymbol{x}^\natural) \leq \gamma \|\boldsymbol{x}^\natural\|$ for any $\gamma \in (0,1)$, with a probability of at least $1 - 8m^{-1}$.

## 3.3 Theoretical results

Given the nonconvex nature of both the objective function and the constraint set in the sparse phase retrieval problem, a thorough theoretical analysis is essential for ensuring the convergence of our algorithm to the ground truth. In this subsection, we provide a comprehensive analysis of the convergence of our algorithm for both noise-free and noisy scenarios.

### 3.3.1 Noise-free case

We begin by the noise-free case, in which each measurement $y_i = |\langle \boldsymbol{a}_i, \boldsymbol{x}^\natural\rangle|^2$. Starting with an initial guess obtained via the sparse spectral initialization method, the following theorem shows that our algorithm exhibits a quadratic convergence rate after at most $\mathcal{O}\big(\log(\|\boldsymbol{x}^\natural\|/x^\natural_{\min})\big)$ iterations.

**Theorem 3.1.** *Let $\{\boldsymbol{a}_i\}_{i=1}^m$ be i.i.d. random vectors distributed as $\mathcal{N}(\boldsymbol{0}, \boldsymbol{I}_n)$, and $\boldsymbol{x}^\natural \in \mathbb{R}^n$ be any signal with $\|\boldsymbol{x}^\natural\|_0 \leq s$. Let $\{\boldsymbol{x}^k\}_{k\geq 1}$ be the sequence generated by Algorithm 1 with the input measurements $y_i = |\langle \boldsymbol{a}_i, \boldsymbol{x}^\natural\rangle|^2$, $i = 1, \ldots, m$, and the initial guess $\boldsymbol{x}^0$ generated by the sparse spectral initialization method mentioned earlier. There exists positive constants $\rho, \eta_1, \eta_2, C_1, C_2, C_3, C_4, C_5$ such that if the stepsize $\eta \in [\eta_1, \eta_2]$ and $m \geq C_1 s^2 \log n$, then with probability at least $1 - (C_2 K + C_3)m^{-1}$, the sequence $\{\boldsymbol{x}^k\}_{k\geq 1}$ converges to the ground truth $\boldsymbol{x}^\natural$ at a quadratic rate after at most $\mathcal{O}\big(\log(\|\boldsymbol{x}^\natural\|/x^\natural_{\min})\big)$ iterations, i.e.,*

$$\mathrm{dist}(\boldsymbol{x}^{k+1}, \boldsymbol{x}^\natural) \leq \rho \cdot \mathrm{dist}^2(\boldsymbol{x}^k, \boldsymbol{x}^\natural), \quad \forall\, k \geq K,$$

*where $K \leq C_4 \log\big(\|\boldsymbol{x}^\natural\|/x^\natural_{\min}\big) + C_5$, and $x^\natural_{\min}$ is the smallest nonzero entry in magnitude of $\boldsymbol{x}^\natural$.*

The proof of Theorem 3.1 is available in Appendix B.2.

*Remark* 3.2. Theorem 3.1 establishes the non-asymptotic quadratic convergence rate of our algorithm as it converges to the ground truth, leading to an iteration complexity of $\mathcal{O}\big(\log(\log(1/\epsilon)) + \log(\|\boldsymbol{x}^\natural\|/x^\natural_{\min})\big)$ for achieving an $\epsilon$-accurate solution. This convergence rate is significantly faster than those of state-of-the-art methods such as ThWF [25], SPARTA [26], and CoPRAM [27], which, as first-order methods, exhibit only linear convergence. Although HTP [28] is a second-order approach, it fails to establish a quadratic convergence rate, and its iteration complexity, $\mathcal{O}\big(\log(\log(n^{s^2})) + \log(\|\boldsymbol{x}^\natural\|/x^\natural_{\min})\big)$, is higher than that of our algorithm.

*Remark* 3.3. It is worth emphasizing that while the superlinear convergence is extensively established for Newton-type methods in existing literature, it often holds only asymptotically: the ratio of the distance to the optimal solution at $(k+1)$-th and $k$-th iterations tends to zero as $k$ goes to infinity. Consequently, the overall iteration complexity cannot be explicitly characterized. This fact highlights the significance of establishing a non-asymptotic superlinear convergence rate.

### 3.3.2 Noisy case

In real-world scenarios, observations are frequently affected by noise. In what follows, we demonstrate the robustness of our proposed algorithm in the presence of noise within phaseless measurements. Building upon [25, 34], we assume that the noisy measurements are given by:

$$y_i = |\langle \boldsymbol{a}_i, \boldsymbol{x}^\natural \rangle|^2 + \epsilon_i, \quad \text{for } i = 1, \ldots, m,$$

where $\epsilon$ represents a vector of stochastic noise that is independent of $\{\boldsymbol{a}_i\}_{i=1}^m$. Throughout this paper, we assume, without loss of generality, that the expected value of $\epsilon$ is $\boldsymbol{0}$.

**Theorem 3.4.** *Let $\{\boldsymbol{a}_i\}_{i=1}^m$ be i.i.d. random vectors distributed as $\mathcal{N}(\boldsymbol{0}, \boldsymbol{I}_n)$, and $\boldsymbol{x}^\natural \in \mathbb{R}^n$ be any signal with $\|\boldsymbol{x}^\natural\|_0 \leq s$. Let $\{\boldsymbol{x}^k\}_{k \geq 1}$ be the sequence generated by Algorithm 1 with noisy input $y_i = |\langle \boldsymbol{a}_i, \boldsymbol{x}^\natural \rangle|^2 + \epsilon_i, i = 1, \ldots, m$. There exists positive constants $\eta_1, \eta_2, C_6, C_7, C_8$, and $\gamma \in (0, 1/8]$, such that if the stepsize $\eta \in [\eta_1, \eta_2]$, $m \geq C_6 s^2 \log n$ and the initial guess $\boldsymbol{x}^0$ obeys $\mathrm{dist}(\boldsymbol{x}^0, \boldsymbol{x}^\natural) \leq \gamma \|\boldsymbol{x}^\natural\|$ with $\|\boldsymbol{x}^0\|_0 \leq s$, then with probability at least $1 - (C_7 K' + C_8) m^{-1}$,*

$$\mathrm{dist}(\boldsymbol{x}^{k+1}, \boldsymbol{x}^\natural) \leq \rho' \cdot \mathrm{dist}(\boldsymbol{x}^k, \boldsymbol{x}^\natural) + \upsilon \|\epsilon\|, \quad \forall 0 \leq k \leq K',$$

*where $\rho' \in (0, 1)$, $\upsilon \in (0, 1)$, and $K'$ is a positive integer.*

The proof of Theorem 3.4 is provided in Appendix B.3. Theorem 3.4 validates the robustness of our algorithm, demonstrating its ability to effectively recover the signal from noisy measurements.

## 4 Experimental Results

In this section, we present a series of numerical experiments designed to validate the efficiency and accuracy of our proposed algorithm. All experiments were conducted on a 2 GHz Intel Core i5 processor with 16 GB of RAM, and all compared methods were implemented using MATLAB.

Unless explicitly specified, the sensing vectors $\{\boldsymbol{a}_i\}_{i=1}^m$ were generated by the standard Gaussian distribution. The true signal $\boldsymbol{x}^\natural$ has $s$ nonzero entries, where the support is selected uniformly from all subsets of $[n]$ with cardinality $s$, and their values are independently generated from the standard Gaussian distribution $\mathcal{N}(0, 1)$. In the case of noisy measurements, we have:

$$y_i = |\langle \boldsymbol{a}_i, \boldsymbol{x}^\natural \rangle|^2 + \sigma \varepsilon_i, \quad \text{for } i = 1, \ldots, m, \tag{14}$$

where $\{\varepsilon_i\}_{i=1}^m$ follow i.i.d standard Gaussian distribution, and $\sigma > 0$ determines the noise level.

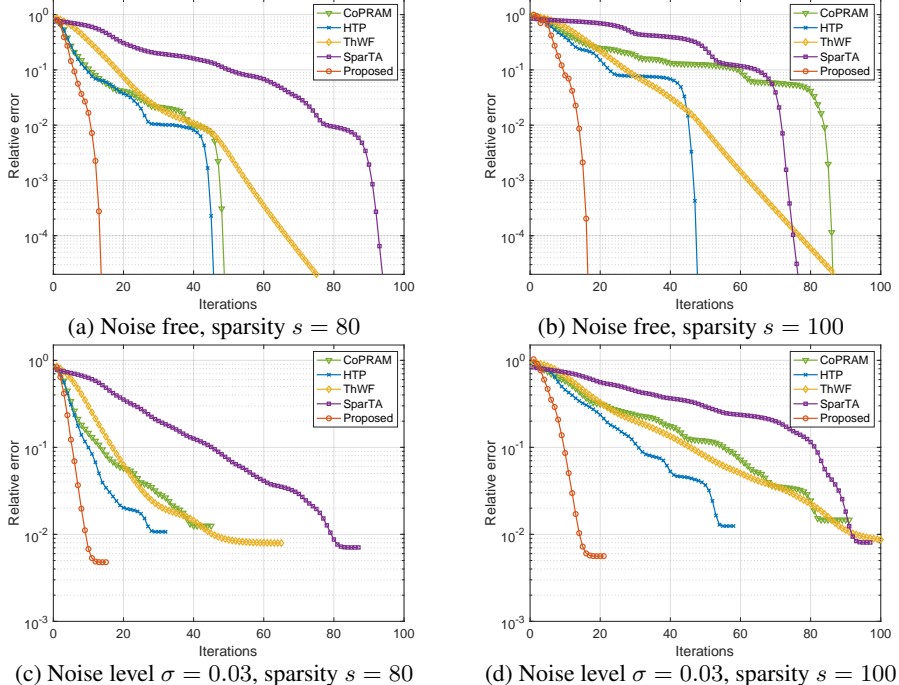

Figure 1: Relative error versus iterations for various algorithms, with fixed signal dimension $n = 5000$ and sample size $m = 3000$. The results represent the average of 100 independent trial runs.

We compare our proposed algorithm with state-of-the-art methods, including ThWF [25], SPARTA [26], CoPRAM [27], and HTP [28]. For ThWF, we set parameters as recommended in [25]. For SPARTA, we set parameters as follows: $\mu = 1$ and $|\mathcal{I}| = \lceil m/6 \rceil$. Both HTP and our algorithm use a step size $\eta$ of 0.95. The maximum number of iterations for each algorithm is 100. The Relative Error (RE) between the estimated signal $\hat{x}$ and the ground truth $x^\natural$ is defined as $\text{RE} := \frac{\text{dist}(\hat{x}, x^\natural)}{\|x^\natural\|}$. A recovery is deemed successful if $\text{RE} < 10^{-3}$. We provide additional experimental results on robustness to noise levels and phase transition with varying sparsity levels in Appendix A.

Figure 1 compares the number of iterations required for convergence across various algorithms. In these experiments, we set the number of measurements to $m = 3000$, the dimension of the true signal to $n = 5000$, and the sparsity levels to $s = 80$ and 100. We consider both noise-free measurements and noisy measurements with a noise level of $\sigma = 0.03$. As depicted in Figure 1, all five algorithms perform well under both noise-free and noisy conditions; however, our algorithm converges with significantly fewer iterations compared to state-of-the-art methods.

Table 2: Comparison of running times (in seconds) for different algorithms in the recovery of signals with sparsity levels of 80 and 100 for both noise-free and noisy scenarios.

| Methods | | ThWF | SPARTA | CoPRAM | HTP | Proposed |
|---|---|---|---|---|---|---|
| Noise free ($\sigma = 0$) | Sparsity 80 | 0.3630 | 1.0059 | 0.9762 | 0.0813 | **0.0530** |
| | Sparsity 100 | 0.6262 | 1.2966 | 3.3326 | 0.2212 | **0.1024** |
| Noisy ($\sigma = 0.03$) | Sparsity 80 | 0.2820 | 1.1082 | 1.3426 | 0.1134 | **0.0803** |
| | Sparsity 100 | 0.4039 | 1.6368 | 4.1006 | 0.2213 | **0.1187** |

Table 2 presents a comparison of the convergence running times for various algorithms, corresponding to the experiments depicted in Figure 1. For noise-free measurements, all algorithms are set to terminate when the iterate satisfies the following condition: $\frac{\text{dist}(x^k, x^\natural)}{\|x^\natural\|} < 10^{-3}$, which indicates a successful recovery. In the case of noisy measurements, the termination criterion is set as $\frac{\text{dist}(x^{k+1}, x^k)}{\|x^k\|} < 10^{-3}$. As evidenced by the results in Table 2, our algorithm consistently outperforms state-of-the-art methods in terms of running time, for both noise-free and noisy cases, highlighting its superior efficiency for sparse phase retrieval applications.

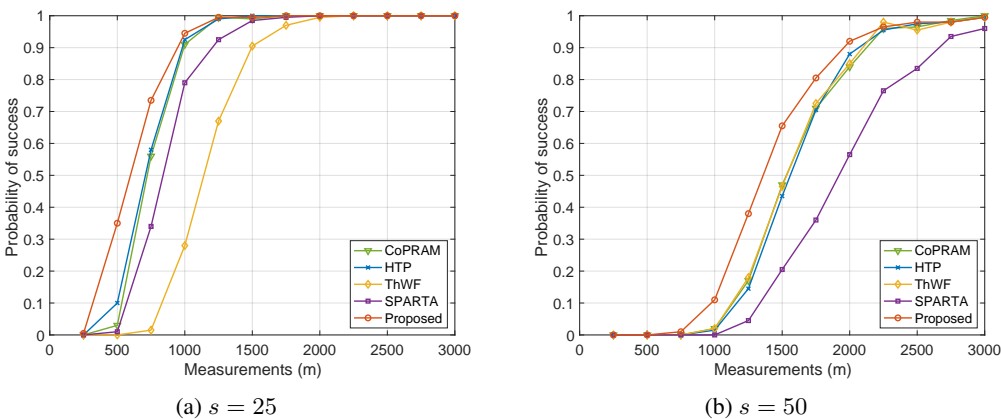

(a) $s = 25$                                (b) $s = 50$

Figure 2: Phase transition performance of various algorithms for signals of dimension $n = 3000$ with sparsity levels $s = 25$ and 50. The results represent the average of 100 independent trial runs.

Figure 2 depicts the phase transitions of different algorithms, with the true signal dimension fixed at $n = 3000$ and sparsity levels set to $s = 25$ and 50. The phase transition graph is generated by evaluating the successful recovery rate of each algorithm over 100 independent trial runs. Figure 2 shows that the probability of successful recovery for each algorithm transitions from zero to one as the sample size $m$ increases. Furthermore, our algorithm consistently outperforms state-of-the-art methods, achieving a higher successful recovery rate across various measurement counts.

In practical applications, natural signals may not be inherently sparse; however, their wavelet coefficients often exhibit sparsity. Figure 3 illustrates the reconstruction performance of a signal from noisy phaseless measurements, where the true signal, with a dimension of 30,000, exhibits sparsity and contains 208 nonzero entries under the wavelet transform, using 20,000 samples. The sampling matrix $A \in \mathbb{R}^{20,000 \times 30,000}$ is constructed from a random Gaussian matrix and an inverse wavelet transform generated using four levels of Daubechies 1 wavelet. The noise level is set to $\sigma = 0.03$. To evaluate recovery accuracy, we use the Peak Signal-to-Noise Ratio (PSNR), defined as:

$$\text{PSNR} = 10 \cdot \log \frac{\text{V}^2}{\text{MSE}},$$

where V represents the maximum fluctuation in the ground truth signal, and MSE denotes the mean squared error of the reconstruction. A higher PSNR value generally indicates better reconstruction quality. As depicted in Figure 3, our proposed algorithm outperforms state-of-the-art methods in terms of both reconstruction time and PSNR. It achieves a higher PSNR while requiring considerably less time for reconstruction. In the experiments, the sparsity level is assumed to be unknown, and the hard thresholding sparsity level is set to 300 for various algorithms.

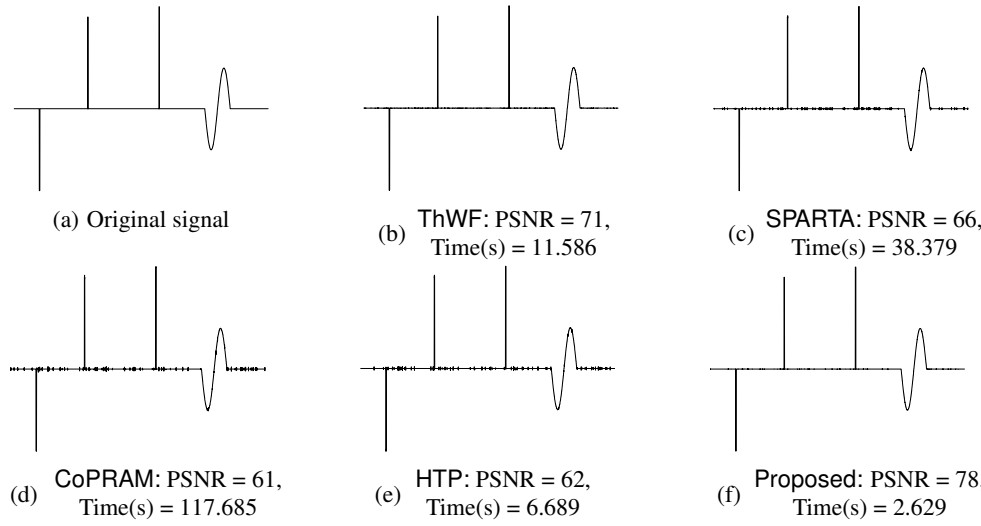

Figure 3: Reconstruction of the signal with a dimension of 30,000 from noisy phaseless measurements by various algorithms. The proposed algorithm requires significantly less time for reconstruction than state-of-the-art methods while preserving the highest PSNR. Time(s) is the running time in seconds.

## 5 Conclusions and Discussions

In this paper, we have introduced an efficient Newton projection-based algorithm for sparse phase retrieval. Our algorithm attains a non-asymptotic quadratic convergence rate while maintaining the same per-iteration computational complexity as popular first-order methods, which exhibit linear convergence limitations. Empirical results have demonstrated a significant improvement in the convergence rate of our algorithm. Furthermore, experiments have revealed that our algorithm excels in attaining a higher success rate for exact signal recovery with noise-free measurements and provides superior signal reconstruction performance when dealing with noisy data.

Finally, we discuss the limitations of our paper, which also serve as potential avenues for future research. Both our algorithm and state-of-the-art methods share the same sample complexity of $\mathcal{O}(s^2 \log n)$ for successful recovery; however, our algorithm requires this complexity in both the initialization and refinement stages, while state-of-the-art methods require $\mathcal{O}(s^2 \log n)$ for initialization and $\mathcal{O}(s \log n/s)$ for refinement. Investigating ways to achieve tighter complexity in our algorithm's refinement stage is a worthwhile pursuit for future studies.

Currently, the initialization stage exhibits a sub-optimal sample complexity of $\mathcal{O}(s^2 \log n)$. A key challenge involves reducing its quadratic dependence on $s$. Recent work [27] attained a complexity of $\mathcal{O}(s \log n)$, closer to the information-theoretic limit, but relied on the strong assumption of power law decay for sparse signals. Developing an initialization method that offers optimal sample complexity for a broader range of sparse signals is an engaging direction for future research.

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
