# OpenReview forum: "A Fast and Provable Algorithm for Sparse Phase Retrieval"
_NeurIPS.cc/2023/Conference — Submitted to NeurIPS 2023_

### Official Review · Reviewer_K7tc · 2023-07-02

**Soundness:** 4 excellent
**Presentation:** 4 excellent
**Contribution:** 4 excellent
**Rating:** 8
**Confidence:** 4

**Summary:**

The authors introduce a novel second-order method for sparse phase retrieval. Compared to previous algorithms, it exhibits faster convergence and better recovery. The method leverages sparsity to reduce the size of the linear system that needs to be solved at each iteration in order to determine the approximate Newton direction (reduced from n^3 to s^3), and a second-order approximation of the intensity-based objective.

**Strengths:**

This paper presents strong results and a theoretical analysis of the algorithm in both the noisy and noise-free case.

**Weaknesses:**

The sample complexity required for initialization and refinement is sub-optimal. The experiments are only on toy data.

**Questions:**

What are the practical implications of the sub-optimal sample complexity required for initializing the algorithm and for the refinement stage? Does it limit the applicability of the method on real-world signals?

Would the authors be able to show experimental comparisons on real-world phase recovery examples?

**Limitations:**

The authors discuss the limitations of their method.

---

> ### Author Rebuttal · Authors · 2023-08-10
>
> > 1. What are the practical implications of the sub-optimal sample complexity required for initializing the algorithm and for the refinement stage? Does it limit the applicability of the method on real-world signals?
>
> **Reply:** Thank you for your insightful questions. The sub-optimal sample complexity required for the refinement stage, as revealed by our extensive experiments, does not adversely impact the practical performance of our algorithm. Indeed, our method demonstrated successful recovery with fewer measurements in numerous numerical experiments. This sub-optimality primarily arises in our theoretical analysis when dealing with the Hessian (please refer to Lemma B.5 for more details).
>
> The sub-optimal sample complexity required for the initialization stage remains an open problem. In the context of sparse phase retrieval, we anticipate a linear dependence of the sample complexity on $s$, whereas our current result shows a quadratic dependence, $\mathcal{O} (s^2 \log n)$. The conditions leading to a linear dependence on $s$ are not yet clear. For example, considering $s = n$ (which reduces the problem to phase retrieval), a linear dependence on $s$ has already been established. An important open question is the lower bound on $s$ that would ensure a linear dependence of the sample complexity.
>
> As to whether this sub-optimality in the initialization stage limits the real-world applicability of our method, the answer is still unclear. Various initialization algorithms have been designed, but the gap of sub-optimality remains. We would like to draw attention to a recent study [R1] which proposes a new initialization method, reducing the sample complexity from $\mathcal{O} (s^2 \log n)$ to $\mathcal{O} (s \bar{s} \log n)$, where $\bar{s}$ represents the stable sparsity of the underlying signal. However, this study does not entirely solve the problem.
>
> In light of your valuable comments, we will include a comprehensive discussion on these topics in our revised manuscript. Your insightful queries will undoubtedly contribute to the thoroughness of our paper.
>
> [R1] J.-F. Cai, J. Li, and J. You, Provable Sample-Efficient Sparse Phase Retrieval Initialized by Truncated Power Method, Inverse Problems, 39(7):075008, 2023.
>
> > 2. Would the authors be able to show experimental comparisons on real-world phase recovery examples?
>
> **Reply:** Thank you for your comment. Our research primarily focuses on the theoretical and algorithmic foundations of sparse phase retrieval problems. The reason for this focus is that we believe a robust theoretical underpinning is crucial for developing reliable and efficient algorithms, which can then be used across a wide range of applications.
>
> As for experimental comparisons on real-world phase recovery examples, we acknowledge the importance of such experiments. However, establishing a real experimental system for phase recovery is not trivial and is outside the scope of our current study. The design and implementation of such an experimental system would require significant resources and expertise in specific application domains, which our team does not currently possess.
>
> We hope that our theoretical contributions will provide a basis for future research, and we look forward to seeing how our results can be applied and validated in real-world settings.

---

> > ### Comment · Reviewer_K7tc · 2023-08-20
> >
> > I thank the authors for their insightful answers to my questions, and for their promise to enrich the manuscript with the discussion on above.
> >
> > I believe this work is an important contribution to the problem of phase retrieval and agree with reviewer CNCT`: this has the potential to become a foundational paper in the field. I also disagree with reviewer kapg's comment regarding this being of interest to few people at the conference.

---

> > > ### Author Response · Authors · 2023-08-21
> > >
> > > We greatly appreciate your insightful comments and the time you've dedicated to reviewing our work. Your recognition of our research's potential impact in the field of phase retrieval is profoundly encouraging.
> > >
> > > Guided by your valuable recommendations, we are committed to enhancing our manuscript by incorporating the discussed points. Once again, we convey our sincerest gratitude for your invaluable feedback and constructive guidance.

---

### Official Review · Reviewer_Sczg · 2023-07-03

**Soundness:** 2 fair
**Presentation:** 3 good
**Contribution:** 2 fair
**Rating:** 5
**Confidence:** 4

**Summary:**

The authors propose a second-order algorithm based in Newton projection for the sparse phase retrieval algorithm. The proposed algorithm is similar to Hard Thresholding Pursuit, where the free variables (i.e. the support) is first identified by a hard thresholding step, followed by an update on the free variables via a Newton projection step.
As is standard for approaches to phase retrieval, the proposed method first performs an initialisation stage to ensure that the initial guess is sufficiently close to the true signal, then applies the proposed second-order method to obtain global convergence. There is are theoretical results proving quadratic convergence for the proposed method.

**Strengths:**

The performance show substantial gains compared to previous methods, moreover, it establishes a quadratic convergence rate.


**Weaknesses:**

This work is incremental compared to HTP of [28]. HTP can already to be interpreted as a second order method. In terms of per-iteration complexity, the proposed method is the same as HTP. The comparison in ‘iteration complexity’ is somewhat unclear because the complexity given in HTP is for exact recovery, whereas the rate given in Table 1 for the proposed method is to obtain accuracy \epsilon — is it just that [28] does not prove a quadratic rate, or do we expect [28] to have worse convergence behaviour in general? Moreover, [28] proves finite convergence for their method, does the proposed method also achieve finite convergence?

In terms of practical performance, the convergence plots show that the proposed method has faster convergence compared to HTP, but the performance for HTP here is worse than the performance reported in [28]. Perhaps it would be useful to replicate the exact experiments in [28] so that a clear comparison can be given? In general, it would be useful to have a discussion on the differences with HTP and an explanation as to why the performance is superior to HTP, given that both are second-order methods. I also had a look at the proof and it is again similar to the proof given in [28], so it would be useful again to have a discussion on the differences and novelty over [28].


**Questions:**

Please clarify on the differences in convergence results between [28] and the proposed method. It is also unclear to me why the proposed method has superior performance both in terms of convergence and in terms of the sparse solutions recovered, when both are second-order methods.

**Limitations:**

Yes

---

> ### Author Rebuttal · Authors · 2023-08-10
>
> > 1. Please clarify on the differences in convergence results between [28] and the proposed method.
>
> **Reply:** We appreciate your constructive suggestion. In response to your query about the differences in convergence results between our method and the one presented in [28], we provide the following clarifications.
>
> In [28], the authors demonstrated that Hard Thresholding Pursuit (HTP) converges to the exact solution within a finite number of steps, specifically, $\mathcal{O} (\log ( s^2 \log n) + \log ( \Vert x^\natural \Vert / x_{\min}^\natural) )$. On the other hand, the convergence rate of our proposed algorithm is $\mathcal{O}(\log (\log (1 / \epsilon) +\log ( \Vert x^\natural \Vert / x_{\min}^\natural ) ))$.
>
> We acknowledge that a direct comparison between these two convergence results is not straightforward. For a more tangible comparison, we could refer to the success criterion used in our research: $\Vert x - x^\natural \Vert / \Vert x^\natural \Vert < 10^{-3}$. Additionally, we often normalize the signal $x^\natural$ such that $\Vert x^\natural \Vert$ equals 1.
>
> Under these conditions, the convergence result of our method simplifies to $\mathcal{O}(\log (\log (10^3) +\log ( 1 / x_{\min}^\natural ) ))$. This could potentially be significantly smaller than the result from [28], $\mathcal{O}( \log ( \log n^{s^2} ) + \log ( 1 / x_{\min}^\natural) )$. This suggests that our proposed method may have a faster convergence rate under the given conditions.
>
> In light of your feedback, we will include a detailed discussion regarding this aspect in our revised manuscript. We believe this will provide a clearer understanding of the comparative advantages of our proposed algorithm.
>
>
> > 2. It is also unclear to me why the proposed method has superior performance both in terms of convergence and in terms of the sparse solutions recovered, when both are second-order methods.
>
> **Reply:** Thank you for your insightful comment. While we cannot provide an exact answer to this question, we have some insights that might explain the observed behavior.
>
> Both our method and the Hard Thresholding Pursuit (HTP) in [28] could indeed be viewed as second-order algorithms. However, HTP does not explicitly construct the Hessian and Newton direction. This difference could be the cause of the superior performance of our algorithm, both in terms of convergence speed and the quality of sparse solutions recovered.
>
> It should be noted that the explicit construction of the Newton direction brings significant challenges in theoretical analysis. This results in a suboptimal sample complexity during the refinement stage of our algorithm's theoretical convergence. To achieve a tighter sample complexity during the refinement stage, a more advanced analytical technique would be needed.
>
> We recognize the value of discussing this matter in greater detail. Therefore, we will include a comprehensive discussion on this topic in our revised manuscript. We believe this will provide a clearer understanding of the comparative advantages of our proposed method.

---

> > ### Comment · Reviewer_Sczg · 2023-08-15
> >
> > Thanks for your response. My score remains unchanged.

---

> > > ### Author Response · Authors · 2023-08-15
> > >
> > > We greatly appreciate the valuable time and effort you've invested in reviewing our work. We understand and respect your decision to maintain the original score. If there are any further comments, queries, or suggestions you wish to convey, please feel free to contact us.

---

### Official Review · Reviewer_kapg · 2023-07-04

**Soundness:** 3 good
**Presentation:** 3 good
**Contribution:** 2 fair
**Rating:** 3
**Confidence:** 5

**Summary:**

This paper focuses on the sparse phase retrieval problem and introduces an efficient second-order algorithm based on Newton‘s method. The algorithm aims to recover sparse signals and offers a quadratic convergence rate while maintaining the same per-iteration computational complexity as first-order methods. Experimental results demonstrate that the proposed algorithm outperforms popular first-order methods in terms of convergence rate and success rate in recovering the true sparse signal.

**Strengths:**

1. The authors' algorithm exhibits a lower complexity per iteration and a higher convergence rate compared to popular first-order methods. It is noteworthy that this is the first algorithm to establish a quadratic convergence rate.
2. The experimental results clearly illustrate the superiority of the proposed algorithm.
3. The paper effectively communicates the motivation behind the development of the second-order algorithm and highlights the complexity reduction achieved by restricting Newton's step to a subset of variables.

**Weaknesses:**

1. The authors mention two prevalent loss functions but do not provide an explanation regarding the difference between these functions in the numerical experiments. It would be beneficial if the authors clearly explain the distinction between the two functions, particularly why the first function is used for initialization and the second one is used in Newton's update.
2. Equation 12 introduces J_{k+1}, which seems to be highly dependent on the choice of S_0, the initial support. This raises concerns about the algorithm's sensitivity to the initial point. It would be valuable for the authors to address this issue and discuss the potential impact of the initial point on the algorithm's performance.
3. Regarding the overall contribution, this paper focuses on approximating the objective function using a quadratic function, which can be limited. Also, this paper may be interested to only a few people attending this conference.



**Questions:**

More extensive experiments would help.  E.g., when designing the experiments for unknown sparsity, it would be better to try different inputs for the sparsity levels. How important is the initialization step?

---

> ### Author Rebuttal · Authors · 2023-08-10
>
> > 1. More extensive experiments would help. E.g., when designing the experiments for unknown sparsity, it would be better to try different inputs for the sparsity levels.
>
> **Reply:** Thank you for your constructive suggestions. We have conducted an additional experiment to address your concerns regarding the handling of unknown sparsity.
>
> In the table below, we consider scenarios with unknown sparsity. We input various sparsity levels into each algorithm and compare the success rates of various algorithms in recovering the signal. In these experiments, the underlying signal has a sparsity of 30, a signal dimension of 3000, and the number of measurements is 2000. We excluded ThWF from the comparison because it does not require input sparsity. Our observations indicate that CoPRAM, HTP, and our proposed algorithm demonstrate greater robustness to changes in input sparsity compared to SPARTA.
>
> | Input sparsity | 10 | 20 | 30 | 50 | 70 | 100 | 150 | 200 | 250 | 300 |
> |----------|----------|----------|----------|----------|----------|----------|----------|----------|-----------|-----------|
> |   CoPRAM    |     0     |     0     |     1     |    1      |    1      |      1    |     0.75     |     0.09     |      0     |     0      |
> |      HTP    |      0    |      0    |     1     |      1    |    1      |     1     |     0.71     |     0.22     |     0.02      |     0.01      |
> |     SPARTA     |     0     |     0     |      1    |     1    |     1     |   0.09       |    0      |    0      |      0     |      0     |
> |     Proposed     |    0      |      0    |    1      |     1     |     1     |    1      |     0.93     |     0.85     |    0.76       |    0.66       |
>
> Once again, we thank the reviewer's insightful feedback. We will include these discussions and experiments in our revised manuscript.
>
> > 2. How important is the initialization step? Equation 12 introduces J_{k+1}, which seems to be highly dependent on the choice of S_0, the initial support. This raises concerns about the algorithm's sensitivity to the initial point. It would be valuable for the authors to address this issue and discuss the potential impact of the initial point on the algorithm's performance.
>
> **Reply:** Thank you for your insightful comment. We agree that the robustness of the algorithm to the initial point is a critical aspect. In response, we have conducted an additional experiment comparing different common initialization methods. **The results are provided in the attached PDF.** These results demonstrate that our algorithm performs well under various initial conditions.
>
> We want to highlight that our theoretical analysis necessitates the initial point to satisfy the condition $\mathrm{dist} (x^0, x^\natural) < \gamma \Vert x^\natural \Vert$ for any $\gamma \in (0,1)$. This condition can be ensured by sparse spectral initialization with a sample complexity of $\mathcal{O} (s^2 \log n)$, with a probability of at least $1 - 8 m^{-1}$. Our theoretical analysis shows that if the initial point fulfills this condition, that is, if the distance is sufficiently close to the underlying signal, the convergence of our algorithm can be guaranteed, without requiring additional conditions on the initial support.
>
> We appreciate your insightful comments and feedback. These discussions and the corresponding experimental results will be included in our revised manuscript.

---

> > ### Comment · Reviewer_kapg · 2023-08-14
> > **Thansk for your response**
> >
> > It is great to see that the proposed algorithm is more robust on the sparsity as long as the number is larger than the exact one.
> >
> > I did not find the pdf for the initialization results. Could you point to me where to find it?

---

> > > ### Author Response · Authors · 2023-08-14
> > >
> > > Thank you for your insightful comments and for recognizing the robustness of our proposed algorithm with respect to sparsity.
> > >
> > > To address your inquiry about the initialization results, they are included in the PDF attached to the "**Author Rebuttal by Authors**" section, located at the beginning of our response.
> > >
> > > The results are presented in Figure 2, where we compare the phase transitions of our algorithm using three different initialization methods: SPI [R1], which we adopted in our initial submission, THI [R2], and HWFI [R3]. Our analysis indicates that our algorithm consistently performs well under each of these initialization methods. Interestingly, we noticed a slightly more robust performance of our algorithm when initialized using SPI and THI compared to HWFI.
> > >
> > > In summary, our algorithm shows robust performance across a range of initialization methods, as demonstrated by our empirical results. Theoretically, our algorithm is guaranteed to converge to the ground truth provided that the initial point meets the condition $\mathrm{dist} (x^0, x^\natural ) < \gamma \Vert x^\natural \Vert$ for any $\gamma \in (0,1)$. This can be ensured under a sample complexity of $\mathcal{O}(s^2 \log n)$ with a probability of at least $1 - 8 m^{-1}$. Importantly, our primary contributions lie in the refinement stage, where we introduce a novel second-order algorithm based on Newton projection and establish non-asymptotic quadratic convergence to the ground truth.
> > >
> > > We greatly appreciate your insightful feedback. These discussions and experimental results will be included in our revised manuscript. We trust this clarifies the location and the details of the initialization results. Please do not hesitate to reach out if any further clarification is required on this or any other matter.
> > >
> > > References:
> > >
> > > [R1] G. Jagatap, and C. Hegde. Sample-efficient algorithms for recovering structured signals from magnitude-only measurements. IEEE Transactions on Information Theory, 65(7):4434– 4456, 2019.
> > >
> > > [R2] T. T. Cai, X. Li, and Z. Ma. Optimal rates of convergence for noisy sparse phase retrieval via thresholded wirtinger flow. The Annals of Statistics, 44(5):2221–2251, 2016.
> > >
> > > [R3] F. Wu, and P. Rebeschini. Hadamard Wirtinger flow for sparse phase retrieval. In International Conference on Artificial Intelligence and Statistics, pp. 982-990, 2021.

---

### Official Review · Reviewer_CNCT · 2023-07-06

**Soundness:** 3 good
**Presentation:** 3 good
**Contribution:** 3 good
**Rating:** 7
**Confidence:** 3

**Summary:**

The work proposes a new algorithm for phase retrieval of sparse signals.
Specifically, it focuses on a faster algorithm targeting quadratic convergence with the same number of measurements that are also needed in other algorithms. A proof of a quadratic convergence rate is established and experments illustrate the benefit also in experiments.

**Strengths:**

The paper focuses on aspects in phase retrieval that are often ignored. In particular, a proofable faster convergence rate has not been the focus of other works so far.
It is well-written and easy to follow.
It may well become a new standard for phase retrieval (or a starting point for other similar algorithms) if other researchers can reproduce the excellent performance.

**Weaknesses:**

The novelty is limited in the sense that second order algorithms are well known. However, adaptation and convergence proof for the phase retrieval setting are indeed novel and interesting.
It is not clear why this subset of existing algorithms has been used for the experiments.

**Questions:**

Are there any existing second order algorithms for phase retrieval (maybe even without convergence proof)?
Why were these specific existing algorithms used for comparsion?
Does increasing the maximum number of iterations in the definition of "successful recovery" change the performance of the various algorithms in Figure 2?

**Limitations:**

The authors present no drawbacks of their method compared to the existing algorithms. In particular, the fact that existing algoritms need fewer measurements for refinement but perfom worse in the phase transition Figure 2 is surprising. Eventually, this is an artifact of the restriction to maximally 100 iterations for success (if the initialization is bad, significantly more iterations might be needed and could still make an algorithm successful, although for a significant computational cost).
It is strange that a faster algorithm is in this sense also more "robust".

---

> ### Author Rebuttal · Authors · 2023-08-09
>
> > 1. Are there any existing second order algorithms for phase retrieval (maybe even without convergence proof)? Why were these specific existing algorithms used for comparsion?
>
> **Reply:** Thank you for your insightful comment. You are correct that there are a few second-order algorithms for phase retrieval and sparse phase retrieval, as outlined in [R1,R2,R3,R4].
>
> Among these, [R1] and [R2] concentrate on phase retrieval, while [R3] and [R4] target sparse phase retrieval. Notably, [R3] does not establish theoretical guarantees for convergence to the true signal, which underscores the necessity and novelty of our work. [R4] introduces a second-order algorithm with theoretical guarantees, which is compared with our algorithm.
>
> The algorithms we chose for comparison in our paper are considered state-of-the-art for sparse phase retrieval and offer theoretical guarantees. Our proposed second-order algorithm not only maintains the same per-iteration computational complexity as popular first-order methods but is also the first to establish a quadratic convergence rate for sparse phase retrieval.
>
> We will ensure to clarify this point in our revised manuscript to highlight the unique contributions of our work over existing second-order methods.
>
> References:
>
> [R1] B. Gao and Z. Xu, “Phaseless recovery using the gauss–newton method,” IEEE Transactions on Signal Processing, vol. 65, no. 22, pp. 5885–5896, 2017.
>
> [R2] C. Ma, X. Liu, and Z. Wen, “Globally convergent levenberg-marquardt method for phase retrieval,” IEEE Transactions on Information Theory, vol. 65, no. 4, pp. 2343–2359, 2018.
>
> [R3] Y. Shechtman, A. Beck, and Y. C. Eldar, “Gespar: Efficient phase retrieval of sparse signals,” IEEE Transactions on signal processing, vol. 62, no. 4, pp. 928–938, 2014.
>
> [R4] J.-F. Cai, J. Li, X. Lu, and J. You, “Sparse signal recovery from phaseless measurements via hard thresholding pursuit,” Applied and Computational Harmonic Analysis, vol. 56, pp. 367–390, 2022.
>
> > 2. Does increasing the maximum number of iterations in the definition of "successful recovery" change the performance of the various algorithms in Figure 2? This is an artifact of the restriction to maximally 100 iterations for success (if the initialization is bad, significantly more iterations might be needed and could still make an algorithm successful, although for a significant computational cost).
>
> **Reply:** Thank you for drawing our attention to this point. We agree that increasing the maximum number of iterations can often improve results, particularly when the number of samples is not sufficiently large. To address this, we updated our experiments by increasing the maximum number of iterations to 1000 for each algorithm and also raised the number of independent trial runs to 200 for averaging.
>
> We observed a slight increase in the probability of successful recovery for each algorithm in the scenario where $s = 50$, while no consistent increase was observed in the case of $s = 25$. It is worth noting that this adjustment to our experimental settings does not alter our original conclusion. **The results are provided in the attached PDF.**
>
> We appreciate your suggestions and will incorporate these updated experimental results in the revised manuscript.
>
> > 3. The authors present no drawbacks of their method compared to the existing algorithms. In particular, the fact that existing algoritms need fewer measurements for refinement but perform worse in the phase transition Figure 2 is surprising.
>
> **Reply:** Thank you for your insightful comment. We would like to clarify that, although theoretically our algorithm requires a larger sample complexity in the refinement stage for successful recovery when compared to other algorithms, this does not necessarily imply that it also needs more measurements in practice. The larger sample complexity arises when establishing Lemma B.5, where we bound a term related to the Hessian—a term not involved in the theoretical analysis of the compared algorithms.
>
> A more advanced technique would be needed in our theoretical analysis to achieve a tighter sample complexity during our algorithm's refinement stage. Additionally, it's important to note that the practical improvement of our algorithm in terms of sample size for successful recovery is not reflected in our theoretical analysis.
>
> We appreciate your suggestion and will include a more detailed discussion on this aspect in our revised manuscript.

---

> > ### Author Response · Authors · 2023-08-12
> > **Thank You for the Increased Review Score**
> >
> > Dear Reviewer,
> >
> > We noticed that you have increased the score for our submission. We would like to express our sincere gratitude for your time and consideration in reviewing our work. We appreciate your positive recognition and are encouraged by it. Please do not hesitate to reach out if you have any other questions.
> >
> > Thank you once again.
> >
> > Best regards,
> > The Authors

---

> > ### Comment · Reviewer_CNCT · 2023-08-16
> >
> > Thank you for the detailed response. Indeed, I did increase my score due to the elaborate answer of all questions raised in my review. I'm looking forward to reading the revised manuscript.

---

> > > ### Author Response · Authors · 2023-08-16
> > >
> > > Thank you for your positive feedback and for recognizing our efforts to address all the questions raised in your review. We greatly appreciate your constructive comments, which have guided us in improving our manuscript.

---

### Author Rebuttal · Authors · 2023-08-10

Dear Reviewers,

We sincerely thank you for dedicating your time to review our manuscript and for your insightful comments. Your feedback has significantly contributed to improving the clarity and overall quality of our paper.

In response to the concerns raised, we have conducted additional experiments and included the results in the attached PDF. We hope that these additional results will address your concerns and strengthen our paper. We look forward to your continued feedback.

---

### Decision · Program_Chairs · 2023-09-21

**Decision:**

Reject

**Comment:**

In this paper, the authors came up with a second-order method (based on suitable Newton projection) for sparse phase retrieval, a well-studied problem that seeks to recover an s-sparse signal from magnitude-only measurements. When the algorithm proposed by the authors is initialized by sparse spectral initialization, it enjoys provable quadratic convergence after it is run for a logarithmic number of iterations. Achieving asymptotic quadratic convergence rates while maintaining the same per-iteration cost as first-order methods has not been shown in prior literature. Nevertheless, some reviewers feel that the algorithmic ideas are not surprising given the large body of work studying the rapid convergence of Newton-type methods and sparse phase retrieval, which I concur. Additionally, the reviewers also raised concerns about the technical contributions of this paper; for instance, the proof strategy is similar to the ones developed in prior works (e.g., the analysis for HTP in [28]), and hence the technical novelty might be inadequate.